# Nisoldipine Inhibits Influenza A Virus Infection by Interfering with Virus Internalization Process

**DOI:** 10.3390/v14122738

**Published:** 2022-12-08

**Authors:** Yingna Huang, Yinyan Li, Zhixuan Chen, Liurong Chen, Jinlong Liang, Chunyu Zhang, Zhengyin Zhang, Jie Yang

**Affiliations:** National Medical Products Administration (NMPA) Key Laboratory for Research and Evaluation of Drug Metabolism, Guangdong Provincial Key Laboratory of New Drug Screening, School of Pharmaceutical Sciences, Southern Medical University, Guangzhou 510515, China

**Keywords:** influenza A virus, SARS-CoV-2, viral entry, nisoldipine, internalization

## Abstract

Influenza virus infections and the continuing spread of severe acute respiratory syndrome coronavirus 2 (SARS-CoV-2) are global public health concerns. As there are limited therapeutic options available in clinical practice, the rapid development of safe, effective and globally available antiviral drugs is crucial. Drug repurposing is a therapeutic strategy used in treatments for newly emerging and re-emerging infectious diseases. It has recently been shown that the voltage-dependent Ca^2+^ channel Cav1.2 is critical for influenza A virus entry, providing a potential target for antiviral strategies. Nisoldipine, a selective Ca^2+^ channel inhibitor, is commonly used in the treatment of hypertension. Here, we assessed the antiviral potential of nisoldipine against the influenza A virus and explored the mechanism of action of this compound. We found that nisoldipine treatment could potently inhibit infection with multiple influenza A virus strains. Mechanistic studies further revealed that nisoldipine impaired the internalization of the influenza virus into host cells. Overall, our findings demonstrate that nisoldipine exerts antiviral effects against influenza A virus infection and could serve as a lead compound in the design and development of new antivirals.

## 1. Introduction

Influenza A virus (IAV) is a major pathogen in humans and various animal species, with epidemic and global pandemic potential. As a consequence, a large number of cases occurring globally every year seriously threaten human health. According to the World Health Organization (WHO), the annual influenza epidemic leads to approximately 3 to 5 million cases of severe illness and about 290,000 to 650,000 respiratory deaths [1]. 

Vaccination is considered the most effective way to protect both human and animal populations against epidemic influenza, but vaccines need to be constantly adapted for the emergence of new circulating strains [2]. Besides the development of efficient vaccines, anti-influenza agents represent an important alternative therapy for both the prophylaxis and treatment of influenza [3,4]. Currently, only three classes of antiviral drugs are approved for use against influenza: the M2 ion channel, neuraminidase and viral RNA polymerase inhibitors. However, M2 inhibitors (amantadine and rimantadine) are obsolete due to widespread resistance and severe side effects [5]. As a result, neuraminidase (NA) inhibitors (oseltamivir, zanamivir, peramivir and laninamivir [6]) and a polymerase acidic protein (PA) inhibitor (baloxavir marboxil [7]) have become the mainstays of antiviral therapy. However, the increasing resistance of the virus to NA and PA inhibitors, due to the high mutagenic capacity of IAV, is a matter of concern [8,9]. Although some innovative drugs have been developed, such as pimodivir [10], the currently available antiviral options for influenza are still limited. Therefore, it is necessary to develop new antiviral approaches to fight influenza, with a particular focus on virus host–cell interactions.

Influenza viruses require the host cellular machinery for the virus life cycle, which can be divided into virus attachment, entry, replication, and budding. Most classic anti-influenza virus drugs act by directly targeting virus-encoded factors. Antiviral strategies aimed at the host cell factors required for the IAV replication process would be advantageous for combating influenza [11,12,13]. Recently, it has been shown that some host cell factors involved in IAV infection, such as GBP7 [14] and SNW1 [15], can serve as novel targets for antiviral drug development.

Nisoldipine is a recognized dihydropyridine calcium channel blocker that specifically affects the L-type voltage-gated calcium channel. It has been widely used in the treatment of hypertension, as it acts as an arterial vasodilator [16]. Clinical studies demonstrated that nisoldipine and its long-acting formulations exhibit good antihypertensive efficacy and tolerability [17]. In recent studies, a relationship between L-type Ca^2+^ channel α1C subunit (Cav1.2) and influenza Hemagglutinin (HA) has been demonstrated, and a possible role for a sialylated voltage-dependent Ca^2+^ channel in infectious entry was illustrated [18,19,20]. Fujioka et al. showed that sialylated voltage-dependent Ca^2+^ channel Cav1.2 as a host cell surface receptor mediates the entry of IAV into host cells [19], providing a target for therapeutic intervention. Furthermore, as nisoldipine is specific for L-type Cav1.2, we wonder if nisoldipine could be used beyond current indications for potential application against IAV infection.

Moreover, drug repurposing strategies and their potential therapeutic effect might be useful for fighting a broad range of human pathogens, especially important for newly emerging viruses such as severe acute respiratory syndrome coronavirus 2 (SARS-CoV-2). Coronavirus disease 2019 (COVID-19), caused by SARS-CoV-2 infections, rapidly spread worldwide, posing a significant threat to global health. As of yet, there is insufficient evidence to show that the available antivirals are sufficiently effective for treating COVID-19. Considering the time-consuming process of developing new drugs, developing effective antivirals from existing licensed drugs is a faster and cheaper pathway.

Here, we assessed the activities of nisoldipine against IAV in vitro and elucidated its mechanism of action. Its combination with an NA inhibitor against IAV infection was also evaluated. We propose that nisoldipine could be a potential candidate for use as an antiviral agent against IAV.

## 2. Materials and Methods

### 2.1. Chemicals and Antibodies

Nisoldipine (T0163), cilnidipine (T0388), nitrendipine (T0119), nimodipine (T0343) and methyl-β-cyclodextrin (MβCD, T4072) with 98% purity were purchased from Target Molecule Corp (Shanghai, China). Chlorpromazine hydrochloride (CPZ, C7010) was purchased from Solarbio (Beijing, China). Ammonium chloride (A801304) was obtained from Macklin (Shanghai, China). EGTA, BAPTA-AM (1,2-bis (o-aminophenoxy) ethane-N, N, N′, N′-tetraacetic acid acetoxymethyl ester) and zanamivir were obtained from Sigma-Aldrich (St. Louis, MO, USA). CL-385319 and D715-2441 with 98% purity were synthetized in our laboratory. Alexa 568-conjugated human transferrin (Tf-568, T23365) was purchased from Invitrogen (Carlsbad, CA, USA). FITC-conjugated cholera toxin beta subunit (CTB-FITC, abs80003) was purchased from Absin (Shanghai, China). The following antibodies were used: influenza A virus PB2 protein antibody (Genetex, GTX125926); influenza A virus NP antibody (Genetex, GTX125989); mouse anti-transferrin antibody (Bioss, bsm-33244M); rabbit anti-caveolin-1 antibody (Bioss, bs-1453R); anti-Cav1.2 antibody (Abcam, ab84814); and GAPDH (glyceraldehyde-3-phosphate dehydrogenase)/β-actin rabbit mAb (Bioss, bs-0061R-2). Anti-rabbit/mouse HRP (horseradish peroxidase)-conjugated secondary antibody (FDR007, FDM007) was obtained from Hangzhou Fdbio Science (Hangzhou, China).

### 2.2. Cells, Viruses and Plasmids

Madin–Darby canine kidney (MDCK) and human embryonic kidney (HEK-293T) cells were obtained from the American Type Culture Collection (ATCC, Manassas, VA USA). All of these cells were cultured in Dulbecco’s Modified Eagle Medium (DMEM) (Gibco Invitrogen) containing 10% fetal bovine serum (FBS, Gibco Invitrogen) and 1% penicillin/streptomycin (Gibco Invitrogen) at 37 °C with 5% CO_2_ incubation. Human lung epithelial (A549) cells (ATCC, Manassas, VA, USA) were grown in RPMI1640 medium supplement with 10% FBS and 1% penicillin/streptomycin. IAV strains including A/Puerto Rico/8/34 (H1N1), A/FM-1/1/47 (H1N1), A/WSN/33 (H1N1), A/PR/8/34 with NA-H274Y and A/Aichi/2/68 (H3N2) were propagated in embryonated chicken eggs at 37 °C for two days. After centrifuging at 2000 rpm for 10 min, the supernatant was stored at −80 °C until use. All experiments with the virus were performed in a BSL-2 laboratory, in accordance with the institutional biosafety operating procedures.

Plasmids pHW2K-NP, pHW2K-PA, pHW2K-PB1, pHW2K-PB2 and pPolI-Fluc (firefly luciferase reporter plasmid) were kindly supplied by Professor Bojian Zheng (University of Hong Kong, Hong Kong, China). The hRluc-TK (Renilla luciferase plasmid) was obtained from Promega (Madison, WI, USA).

### 2.3. Cytotoxicity Studies

The cell viability of the target cells incubated with increasing concentrations of compounds was measured by 3-(4,5-dimethylthiazol-2-yl)-2,5-diphenyltetrazolium bromide (MTT, Sigma-Aldrich) assay after 48 h of treatment. MTT was then added to the cells at a final concentration of 0.5 mg/mL, and the mixture was incubated for 4 h (37 °C, 5% CO_2_). After incubation, formazan crystals were dissolved in 150 μL of dimethyl sulfoxide (DMSO). The absorbance at 570 nm was measured with a microplate reader (GENios Pro, TECAN, Bedford, MA, USA).

### 2.4. Antiviral Assay and Microscopy

The MDCK cells were seeded in 96-well plates at a density of 2 × 10^4^ cells/well and incubated overnight at 37 °C and 5% CO_2_. Cell monolayers were pre-treated with the compounds prior to inoculation with IAV at 100 TCID50 by adding compounds at a final concentration of 10 μM to the medium followed by incubation for 1 h at 37 °C. After 48 h post-infection, virus-induced cytopathic effects (CPE) were observed by microscopy, and the antiviral activity of the compounds was determined by MTT-based assay as described previously [21]. Half maximal inhibitory concentration (IC_50_) was calculated using Calcusyn computer software [21].

### 2.5. Plaque Reduction Assay

The MDCK cells were pre-treated with various concentrations of the compounds for 1 h followed by infection with influenza A/WSN/33 virus at a multiplicity of infection (MOI) of 0.01. The virus-infected cells were cultured by a 2 × DMEM (Sigma, St. Louis, MO, USA) mixed medium containing 1.5 μg/mL TPCK-trypsin (Sigma, St. Louis, MO, USA), 2% microcrystalline cellulose and the compounds for 3 days. The cells were then fixed with 4% paraformaldehyde for 20 min. After discarding the medium, the cells were subsequently stained with 2% crystal violet. The number of infectious particles was determined using a standard plaque assay. The inhibition effects of the compounds depended on the number of plaques.

### 2.6. Western Blotting and Quantitative Real-Time PCR

To obtain the total protein, the cells were lysed using radio-immunoprecipitation assay (RIPA). For Western blotting, the total protein was quantified by Bradford assay, and denatured by boiling at 105 °C. Equal amounts of protein were separated on SDS-PAGE gels and transferred to a polyvinylidene fluoride (PVDF) membrane using an electro-blotting apparatus (Bio-Rad, Hercules, CA, USA). The membrane was blocked for 60 min at room temperature with 5% bovine serum albumin (BSA) and incubated overnight at 4 °C with the primary antibody. The membrane was then incubated with an HRP-conjugated secondary antibody for 1 h at room temperature (RT). The bound antibodies were detected using an enhanced chemiluminescence substrate (Bio-rad, United States). Images were obtained using the FluorChem E System (Protein-sample, Santa Clara, CA, USA) and analyzed using ImageJ software (NIH, Bethesda, MD, USA).

The cells were lysed using an RNA isolation kit (Foregene, Chengdu, China). The mRNA expression level was detected with qRT-PCR as previously reported [22]. The relative expression of the viral gene was determined with LightCycler 480 instrument software [22] using a classical 2^-ΔΔCT^ method, with the cellular GAPDH gene serving as the internal control. The PrimeScript™ RT Reagent Kit (TaKaRa, Beijing, China) and the GoTaq^®^ qPCR Master Mix (Promega, Madison, WI, USA) were used for the RT and qRT-PCR experiments, respectively. The qRT-PCR primer sequences are available upon request.

### 2.7. Time of Addition Study

A549 cells were infected with A/WSN/33 viruses (MOI = 0.1) for 1 h and then incubated in the presence of the compound at 20 μM at different time intervals (0–2, 2–5, 5–8, 8–10 and 0–10 h), as previously reported [23]. After 10 h post infection, the cells were collected, and the NP protein levels in the cells were detected using Western blotting, as described above. All assays were performed in three replicates.

### 2.8. Pseudovirus-Based Entry Inhibition Assays

Pseudovirus-based entry inhibition assays, including H5N1 IAV and VSV-G, were generated in our laboratory as described previously [24]. In brief, 293T cells were transfected with HIV backbone plasmid (pNL4-3.luc.R-E-), and the plasmid expressed the viruses’ respective envelope glycoprotein (Qinghai-HA, Xinjiang-HA, Anhui-HA, HongKong-HA, IAV NA or VSV-G). Supernatants were harvested at 48 h post-transfection and produced pseudotype viruses (PsVs). The PsV titers and the inhibition activities of the compounds were measured using a luciferase assay (Promega, Madison, WI, USA). Briefly, the compounds and the PsVs were mixed at 37 °C for 30 min. The cells were subsequently incubated with this mixture. After a further 48 h of incubation, the cells were lysed for luciferase assays. The relative luciferase activity was measured using an Ultra 384 luminometer (GENiosPro, TECAN, Bedford, MA, USA).

### 2.9. Mini-Replicon Assay

Viral polymerase activity was assessed using the mini-replicon system with viral polymerase expression vectors (pHW2K-NP, pHW2K-PA, pHW2K-PB1 and pHW2K-PB2, in a 1:1:1:1 ratio) and two luciferase reporter genes, as described previously [25]. Renilla luciferase expression plasmid was used as a reference gene for the normalization of firefly luciferase reporter gene expression in the transfected cells. The luciferase activity of the cell lysates was determined using a luciferase reporter gene detection kit (Promega, Madison, WI, USA).

### 2.10. IAV Binding, Internalization and Membrane Fusion Assay

The A549 cells were treated with nisoldipine (20 μM) or vehicle (DMSO) at 37 °C for 2 h, followed by infection with A/WSN/33 virus at an MOI of 5 in the presence of ammonium chloride (50 mM) and incubated at 37 °C for 2 h (ammonium chloride was added to prevent fusion). After three washes with ice-cold PBS, the relative level of bound viral particles was quantified via qRT-PCR, as described above.

For the virus binding assay, the A549 cells were pre-treated with 20 μM nisoldipine or DMSO at 37 °C for 3 h and incubated with A/WSN/33 virus (MOI of 5) at 4 °C for an additional 1 h. After three washes with ice-cold PBS, the cell lysates were prepared with RIPA (Fdbio Science) and subjected to Western blotting assays with a rabbit anti-NP pAb.

For the virus internalization assay, the A549 cells were infected with A/WSN/33 (H1N1) virus (MOI of 5) for 1 h at 4 °C. The infected cells were washed five times with ice-cold PBS (pH 7.2) or PBS-HCl (pH 1.3). The cell lysates were then prepared with RIPA and subjected to Western blotting assays with a rabbit anti-NP pAb.

The A549 cells were treated with nisoldipine or vehicle for 3 h at 37 °C prior to infection with A/WSN/33 virus (MOI of 5) on ice at 4 °C for 1 h. The temperature was shifted to 37 °C to allow internalization. The cells were washed in ice-cold PBS-HCl (pH 1.3) five times to remove the uninternalized virions. At the indicated time points post-infection, the cell lysates were subjected to Western blotting with a rabbit anti-NP pAb.

### 2.11. Transferrin Absorption Test

The A549 cells were treated with different concentrations of nisoldipine for 3 h, 2 mL of transferrin (15 µg/mL) was added, and they were then incubated in the cell culture incubator for 30 min. The cell lysates were prepared and subjected to Western blotting assays with a mouse anti-transferrin mAb.

### 2.12. Transmission Electron Microscopy

As it was difficult to detect virus particles at low doses using electron microscopy, the cells were infected at a high MOI. The MDCK cells were treated with nisoldipine as described above. Subsequently, the MDCK cells were incubated with A/WSN/33 virus at an MOI of 10 for 1 h at 4 °C and then transferred to 37 °C and incubated for 30 min, allowing for virus internalization. Next, the cells were fixed with 0.1% glutaraldehyde and 2% paraformaldehyde (PFA) in PBS (pH 7.4) for 1 h at 4 °C. After fixation, the samples were post-fixed with 1 mL of 2% osmium tetroxide for 1 h at 4 °C. The samples were then dehydrated through a graded series of ethanol and embedded in Epon (TED Pella, Redding, CA, USA). Ultrathin sections (80 nm) of the embedded cells were prepared, stained with uranyl acetate and lead citrate, and observed using a transmission electron microscope (Hitachi H-7500, Hitachi High-Technologies Corporation , Nagoya, Japan).

### 2.13. siRNA Knockdown of Cav1.2

The A549 cells were transfected with siRNA targeting Cav1.2 (5′-UCUGAAGAUCCUGUCAGGCAAUTT-3′) or negative control siRNA (GenePharma, Shanghai, China) at a concentration of 30 nM using the Endofectin™ Max transfection reagent (Invitrogen, Carlsbad, CA, USA). At 24 h post-transfection, the knockdown efficiency was verified by Western blotting with a mouse anti-Cav1.2 mAb. The siRNA-treated A549 cells were infected with A/WSN/33 virus (MOI = 0.1) for 1 h at 37 °C. The inhibitory effects of Cav1.2 siRNA on virus replication at indicated points post-infection were determined by Western blotting. All assays were performed in three replicates.

### 2.14. Assessment of Combination Treatment In Vitro

The IC_50_ values of nisoldipine and zanamivir in the MDCK cells were determined using an MTT assay and then used to generate a fixed ratio for the combination studies. During the different dose combinations, the ration of IC_50_ as nisoldipine:zanamivir was 7:1, 3:1, 1:1, 1:3 and 1:7. The MDCK cells were infected with A/WSN/33 virus at 37 °C; then, the drug mixtures were incubated at above the ratio of IC_50_ values. A fractional inhibitory concentration index (FICI) was used to confirm the synergistic effects: FICI = [(IC_50_ of nisoldipine in combination)/(IC_50_ of nisoldipine alone)] + [(IC_50_ of zanamivir in combination)/(IC_50_ of zanamivir alone)]. FICI < 0.5 was interpreted as synergy [26].

### 2.15. Fluorescence Confocal Assays

The A549 cells were plated on 35 mm confocal dishes and cultured for 24 h until they reached 80% confluence. These cells were then initially pre-treated with CPZ (20 μM), MβCD (20 μM) and nisoldipine (20 μM) for 3 h at 37 °C. CPZ acts as an inhibitor of clathrin-mediated endocytosis. MβCD is an inhibitor of caveolin-mediated endocytosis. Next, the A549 cells were labeled with endocytic markers sorted into different vesicular pathways (25 µg/mL Tf-568 or 2 µg/mL CTB-FITC) and incubated at 4 °C for 1 h. The A549 cells were then incubated at 37 °C for 15 min. Finally, the A549 cells were fixed for observation. Images were acquired using a Zeiss LSM 800 Confocal Laser Scanning Microscope with Airyscan. All assays were performed in three replicates.

### 2.16. Statistical Analysis

The independent experiment was performed at least three times. The results are shown as mean ± standard deviation (SD). All statistical analyses of the data were carried out using GraphPad 5.0 Prism software. Two groups were analyzed using statistical methods, including Student’s t-test, and other groups by one-way ANOVA with or without Tukey–Kramer multiple comparisons. In all cases, a *p* value < 0.05 was regarded as statistically significant and marked with an asterisk (*). All results are representative of three replicate experiments (ns, no significance).

## 3. Results

### 3.1. In Vitro Anti-Influenza Activity of Nisoldipine

We previously performed a screen of a compound library of 361 ion channel inhibitors (L2300 Ion Channel Inhibitor Library, Target Molecule Corp.) and identified candidate compounds with anti-IAV activity (inhibition rate > 60%) without significant cytotoxicity at a concentration of 10 μM. Among the candidate drugs identified in the primary screening, calcium channel blockers belonging to the dihydropyridines class displayed better anti-influenza activity. We compared the anti-IAV activity of four represented drugs of dihydropyridine calcium channel blockers, nitrendipine, nimodipine, cilnidipine and nisoldipine, which have been extensively evaluated for clinical safety and have potency for repurposed drug application. Zanamivir was selected as the positive drug control, with an IC_50_ of 1.26 ± 0.30 nM (Table 1). The results showed that the four candidate drugs had different inhibitory effects against influenza A virus (A/Puerto Rico/8/34): the IC_50_ were 7.26 ± 1.89, 12.83 ± 1.72, 14.26 ± 2.66 and 4.74 ± 0.76 μM (Table 1).

Among them, nisoldipine displayed the most powerful inhibition effect on IAV infection without causing cytotoxicity (CC_50_ > 200 μM), with a selection index (SI) greater than 42.19. Fujioka et al. reported that the voltage-gated Ca^2+^ channel Cav1.2 was critical for IAV infection [19]. Our results consistently demonstrated that nisoldipine specific for L-type Cav1.2 inhibited the infection of IAV in vitro. Furthermore, nisoldipine exhibited broad-spectrum inhibition against multiple subtypes of IAV strains (Table 2), including A/Puerto Rico/8/34 (H1N1), A/FM-1/1/47 (H1N1), A/Aichi/2/68 (H3N2), A/WSN/33 (H1N1), and A/PR/8/34 with NA-H274Y, with IC_50_ values of 4.74 ± 0.76, 6.96 ± 0.78, 5.68 ± 0.61, 4.47 ± 0.25 and 5.94 ± 0.55 μM, respectively. Based on the significant inhibition activity of nisoldipine against the A/WSN/33 virus strain, we performed subsequent experiments with the virus strain.

The chemical structure of nisoldipine is shown in Figure 1A. The results of CPE and plaque assays further revealed that nisoldipine could effectively inhibit the replication of IAV in MDCK cells (Figure 1B,C). At the same time, a reduction in progeny virion of the nisoldipine-treated supernatant was detected using a plaque assay (Figure 1D). Western blotting and qRT-PCR were used to analyze the expression of the influenza viral protein and gene. Western blotting showed that nisoldipine significantly inhibited the expression of IAV proteins NP and PB2 in a dose-dependent manner (Figure 1E). Similarly, the results of the qRT-PCR assay demonstrated that nisoldipine decreased NP mRNA expression (Figure 1F). Concomitantly, nisoldipine also interfered with IAV replication in the A549 cells (Figure 1G,H), suggesting that its antiviral effect has no cell specificity. Taken together, these results demonstrate that nisoldipine is a potential anti-influenza drug and that further research would be of value.

### 3.2. Nisoldipine Inhibits Viral Entry

We subsequently investigated the inhibitory mechanism of nisoldipine against IAV infection. The IAV life cycle was mainly around 8–10 h, which can be divided into the following steps: virus entry (0–2 h); viral transcription and replication (2–5 h); viral assembly (5–8 h); and viral release (8–10 h) [27]. To compare the inhibition effect of nisoldipine during the different stages of infection, the A549 cells were treated with nisoldipine at indicated time intervals during one viral life cycle. The results showed that nisoldipine could greatly lessen the expression of the viral NP protein at the 0–2 h time interval, suggesting that nisoldipine acts in the early steps of the viral life cycle (Figure 2A,B).

To verify whether nisoldipine interferes with virus entry, we generated four pseudoviruses in 293T cells with an HIV-1 backbone, expressing HA of four homologous clades of H5N1 IAV (A/Qinghai/59/2005, A/Xinjiang/1/2006, A/Anhui/1/2005, and A/Hong Kong/156/1997) [24]. As depicted in Table 3, it was found that nisoldipine could also significantly inhibit infection with all four pseudoviruses, implying that nisoldipine has effect on viral entry. Similar to CL−385319 [28], nisoldipine could dose-dependently inhibit H5N1 PsV infection but showed much less efficiency on vesicular stomatitis virus G (VSV-G) PsV (Figure 2C).

### 3.3. Nisoldipine Exerts Its Antiviral Effect by Interfering with Influenza A Virus Internalization

To further investigate the precise stage at which nisoldipine affects IAV entry, we studied the effects of nisoldipine on virus–receptor binding, internalization and membrane fusion.

To prevent membrane fusion and virus entry, we added ammonium chloride to allow for the investigation of pre-fusion events, including binding and internalization, as described previously [29,30]. The A549 cells were incubated with nisoldipine or vehicle (DMSO) and then infected with A/WSN/33 in the presence of ammonium chloride (50 mM). At 24 h post-infection, the intracellular level of vRNA was measured with qRT-PCR. Our data show that vRNA in the cells pre-treated with nisoldipine was remarkably reduced, suggesting that nisoldipine acted at the pre-fusion stage of IAV (Figure 3A).

To evaluate the effect of nisoldipine on viral binding, the A549 cells were incubated with nisoldipine or vehicle for 3 h at 37 °C, and then infected with A/WSN/33 (MOI = 5) at 4 °C for 1 h, allowing for virus attachment but preventing virus internalization. As shown in Figure 3B, there was no significant difference in NP protein level observed between the nisoldipine-treated and control cells, suggesting that nisoldipine does not disturb viral binding. After binding to their specific receptor on the host cell surface, IAV is taken up into the cells through the endocytosis pathway. Thus, we investigated the potential effect of nisoldipine on viral internalization. The cells were incubated with A/WSN/33 (MOI = 5) at 4 °C for 1 h, washed five times with a neutral wash (ice-cold PBS, pH 7.2) or an acidic wash (ice-cold PBS-HCl, pH 1.3), and the amount of internalized virus particles was then determined by Western blotting with a rabbit NP pAb. As shown in Figure 3C, the expression of NP was completely absent from the infected cells washed with ice-cold PBS-HCl (pH 1.3), whereas washing with PBS could not remove the virions from the cell surface. Therefore, we chose PBS-HCl (pH 1.3) to remove the unbound viruses and kept the virions that were internalized into the cells. Next, the compound-treated A549 cells, as described above, were infected with A/WSN/33 (MOI = 5) for 1 h at 4 °C. The cells were subsequently transferred to 37 °C for 30 min to allow for internalization. At 15, 30, 45 or 60 min post-infection, the cells were washed with ice-cold PBS-HCl (pH 1.3) five times and lysed for the detection of viral NP protein using Western blotting. Protein expression in the A549 cells after nisoldipine treatment decreased at the same time as the cells treated with DMSO (Figure 3D). Therefore, it was concluded that treatment with nisoldipine could interfere with the internalization process of IAV.

### 3.4. The Effect of Nisoldipine on Clathrin-Mediated or Clathrin-Independent Endocytic Pathway

Influenza virus exploits multiple endocytic pathways for infection. Clathrin-mediated endocytosis may be the most common endocytic pathway and a major internalization pathway for IAV entry into host cells [31]. Using transferrin as a specific marker for clathrin-mediated endocytosis, we further explored the molecular mechanisms by which nisoldipine affected IAV internalization. The internalization of transferrin was assessed upon the treatment of A549 cells with nisoldipine, and as expected, it was found to be markedly reduced (Figure 4A). These results revealed that nisoldipine inhibited the clathrin-associated uptake of transferrin in a concentration dependent manner. To ensure the validity of these data, a transmission electron microscopic (TEM) experiment was undertaken to observe the early stage of IAV infection. At the early stage of infection, the majority of IAV particles were observed to be trapped at the cell surface and partially invaginated, and then, the internalized virions were delivered to an endosome-like structure, suggesting that the entry of IAV into cells could be mediated by endocytosis. Subsequently, we investigated the effect of nisoldipine on the internalization of IAV into the cells. As shown in Figure 4B, the number of clathrin-coated vesicles in the compound-treated A549 cells was less than that in the control cells, suggesting that the entry of IAV into the cells was inhibited by nisoldipine.

Although clathrin-mediated endocytosis plays an important role in the entry of IAV, other endocytic pathways also hinder the infection of IAV. Caveolae have been implicated in the endocytosis of IAV [32]. The A549 cells pre-treated with nisoldipine at different concentrations were inoculated with IAV (MOI = 5). The expression of clathrin-1 protein involved in caveolin-mediated endocytosis was analyzed with Western blotting at 24 h post-infection and compared with the control group. The results clearly demonstrated that IAV utilized caveolin-mediated endocytosis to internalize into host cells as previously reported [33], and the endocytosis pathway was effectively blocked under our experimental conditions (Figure 4C).

To confirm the above result, we employed endocytic marker internalization assays with the endocytic pathway inhibitors (CPZ or MβCD) as a positive control. For the detection of clathrin and caveolin-mediated endocytosis, the A549 cells with a presence or absence of nisoldipine were labeled with different endocytic markers (25 µg/mL Tf-568 or 2 µg/mL CTB-FITC). According to Figure 4D, transferrin uptake was decreased when the cells were pre-treated with nisoldipine compared with the control, revealing that nisoldipine successfully inhibited clathrin-mediated endocytosis. Similar to the results of the transferrin uptake assays, less immunostaining was observed in nisoldipine- or MβCD-treated A549 cells compared with the control A549 cells, indicating that nisoldipine successfully inhibited caveolin-mediated endocytosis (Figure 4E).

To exclude whether nisoldipine suppressed the influenza virus through other mechanisms, we performed three experiments: traditional hemagglutination (HA) inhibition assay, mini-replicon assay and NA inhibition assay. No significant inhibition effect was observed on red blood cells from hemagglutination nor NA enzyme activity, suggesting that nisoldipine is unable to inhibit the function of viral surface glycoproteins HA or NA (Appendix A). Following internalization, the viral membrane must be fused with the cellular membrane, allowing for viral RNP release into the cytoplasm. We further investigated whether the inhibitory mechanism of nisoldipine was involved in the transcription and replication of the IAV genome with a mini-replicon assay. D715-2441 was previously reported as a viral PB2 inhibitor [27]. In contrast with D715-2441, nisoldipine did not affect influenza viral polymerase activity (Appendix A), indicating that nisoldipine has no inhibitory effect on viral transcription/replication. In conclusion, nisoldipine played an antiviral role by inhibiting viral endocytosis.

### 3.5. Nisoldipine Inhibits IAV Infection by Reducing Cellular Ca^2+^ Uptake

As calcium channels are the target of nisoldipine, we determined whether the anti-viral effect of nisoldipine is involved in a reduction in the level of intracellular Ca^2+^. The A549 cells were treated with BAPTA-AM or EGTA to determine the changes in both intra and extracellular Ca^2+^. After incubation with A/WSN/33 (MOI = 0.1), the intracellular level of IAV RNA was extracted at 3, 6, 9, 12 and 24 h post-infection, respectively, and measured using qRT-PCR. As shown in Figure 5A,B, intracellular and extracellular Ca^2+^ were chelated with BAPTA (1,2-bis (o-aminophenoxy) ethane-N, N, N′, N′-tetraacetic acid) and EGTA, respectively, while IAV replication was dramatically impaired. The results showed that an increased intracellular Ca^2+^ level was beneficial for IAV infection and that nisoldipine inhibited IAV infection by reducing the intracellular Ca^2+^ level.

To explore the relationship between the calcium channel and virus infection, we further investigated IAV-induced Ca^2+^ influx. We used an intracellular calcium-sensitive dye (BBcellProbeTMF3) to measure the intracellular Ca^2+^ level during IAV infection. We found that IAV infection increased Ca^2+^ influx over time, which was inhibited by treatment with nisoldipine, indicating that there is an association between the antiviral activity of nisoldipine and IAV-induced Ca^2+^ influx (Figure 5C). Furthermore, the results demonstrated that a disruption of the calcium channel led to reduced IAV infection. Thus, nisoldipine inhibited IAV infection by reducing virus-induced Ca^2+^ influx.

### 3.6. Effect of siRNA Knockdown of Cav1.2 on the Replication of IAV

Nisoldipine as a calcium channel blocker of the dihydropyridine subclass is specific to L-type Cav1.2; therefore, Cav1.2 was chosen for further analysis. The A549 cells were transfected with siRNA against Cav1.2 or control siRNA for 24 h and then infected with A/WSN/33 virus (MOI = 0.1) for 1 h at 37 °C, followed by an analysis of the intracellular viral NP level at the indicated time points. The viability of the cells was not affected after the knockdown of Cav1.2 determined by MTT assay (Figure 5D). In the A549 cells, the protein level of Cav1.2 was significantly reduced by its specific siRNA interference (Figure 5E). These data further demonstrate that Cav1.2 positively regulates IAV replication. Moreover, the Cav1.2 calcium channel may be an important target for the inhibition of IAV.

### 3.7. In Vitro Synergistic Anti-Influenza Activity of Nisoldipine with NAI

NA inhibitors, which selectively inhibit NA and then prevent the release of viral particles from infected cells, are recommended for the prevention and treatment of seasonal and pandemic influenza. However, the emergence of resistant strains limits the clinical utility of existing NA inhibitors [34,35]. Zanamivir, as the first of a new class of selective NA inhibitors, is widely used for the treatment of influenza caused by influenza A and B viruses. As the antiviral mechanisms of nisoldipine are different from NA inhibitors, we further explored the antiviral effects of nisoldipine combined with zanamivir against IAV in vitro. We conducted MTT assays on MDCK cells using the IC_50_ concentrations for each drug singly or a combination of nisoldipine and zanamivir. To ascertain an appropriate synergistic combination ratio for nisoldipine and zanamivir, the IC_50_ against A/WSN/33 was determined via an MTT-based assay, as described above. When the FICI < 0.5, it was interpreted as synergy. The results indicated that nisoldipine showed significant synergy with zanamivir against the tested IAV strain, with FICI values of 0.49 (Table 4), indicating that the combination of nisoldipine and zanamivir had a synergistic inhibitory effect on viral replication. These results suggested that nisoldipine might be a potential option in fighting drug-resistant influenza virus.

## 4. Discussion

Dihydropyridine and non-dihydropyridine calcium channel blockers (CCBs) have been widely used in the treatment of hypertension for many years. This latter group (verapamil and diltiazem) has been shown to inhibit IAV infection (Nugent and Shanley, 1984) [19,36], suggesting that the voltage-dependent Ca^2+^ channel is involved in IAV entry and subsequent infection. Therefore, this article exclusively focuses on the effects of dihydropyridine CCBs against IAV infection. Nisoldipine specifically blocks the L-type Cav1.2 channel. Nimodipine and nitrendipine are L-type calcium channel blockers, and cilnidipine blocks both L- and N-type calcium channels. While all these classical dihydropyridine calcium channel blockers inhibited IAV infection, nisoldipine exhibited excellent antiviral activity compared with that of the others. The cell cytotoxicity results suggested that nisoldipine had no significant cytotoxicity to the target cells in the studied concentration range. The anti-IAV activity data indicated that nisoldipine exhibited strong antiviral effects against multiple IAV subtypes, particularly against the oseltamivir-resistant influenza A (H1N1) virus with an H274Y mutation in NA. Evidence has accumulated that the H274Y NA mutation results in a reduction in oseltamivir susceptibility [37]. The anti-influenza virus activity of nisoldipine was further confirmed using qRT-PCR and WB analysis. Therefore, nisoldipine possesses a relatively broad-spectrum antiviral effect and has the potential to fight emerging drug resistance.

Encouraged by the above results, we next investigated the anti-influenza action mechanism of nisoldipine. The pre-treatment of A549 cells with 20 μM of nisoldipine led to a significant suppression of viral NP expression at the early time points after IAV infection, indicating that nisoldipine blocked an early step in the influenza viral life cycle. With IAV-pseudotyped and minigenome models, we found that nisoldipine did not affect viral RNA replication but interfered with virus entry. Furthermore, nisoldipine could not inhibit either the function of viral surface glycoprotein HA or the viral release. Thus, these data clearly showed that the antiviral mechanism of nisoldipine was most likely involved in the host factors rather than targeting the virus itself. Unlike targeting viral proteins, targeting the host factors speculatively possesses a lower likelihood of drug resistance. Our findings are highly valuable for developing nisoldipine as an exceedingly promising candidate for host-directed therapy to counteract virus infection.

A critical step in the infection cycle is virus entry. Recent investigations have indicated that the engagement of sialic acids linked to the Ca^2+^ channel Cav1.2 could bind HA and facilitates IAV entry [19], which is consistent with our results in that the Cav1.2 channel blocker nisoldipine could effectively block IAV entry. We therefore considered how it acts on the entry process of IAV. IAV enters cells through multiple endocytic pathways, and the majority of the virus particles enter cells through clathrin-mediated endocytosis [38]. Here, we demonstrated that nisoldipine affected transferrin uptake when transferrin was added to the cells, suggesting the possible inhibition effect of nisoldipine in clathrin-mediated endocytosis. Virus internalization was further investigated using microscopy analysis and was consistent with the above results. These observations suggest that nisoldipine severely inhibited the clathrin-mediated entry of the influenza virus. In addition, we showed that nisoldipine also inhibited IAV-infected cells through caveolin-mediated endocytosis. Based on these observations, we regarded that the anti-influenza viral efficacy of nisoldipine was attributable to blocking virus entry through a clathrin- and caveolin-independent pathway. However, in virus internalization, following a rapid release of coated vesicles, the internalized vesicles, termed early and late endosomes, usually go through complex cellular machinery [39]. Moreover, different endocytosis pathways may exhibit different intracellular trafficking routes [40]. Therefore, our current work has implications for the basic process of endocytosis. We found that nisoldipine led to a disruption of the IAV entry pathway by blocking virion endocytosis, which provided evidence of the inhibition mechanism of nisoldipine against IAV infection. However, whether nisoldipine correlated with impaired influenza virus trafficking to late endosomes needs further investigation.

To initiate infection, diverse viruses exploit endocytosis for entry to target cells. As reported in recently published studies, SARS-CoV-2 might deposit its RNA genomes into the host cytoplasm through endocytosis [41,42]. In the case of Ebola or Zika virus infection, viral entry is reported to require endocytic uptake [43,44]. To examine whether nisoldipine exerts antiviral activity against other enveloped viruses, we compared the results of virus entry mediated by the glycoproteins of the influenza virus, VSV-G and SARS-CoV-2. Entry inhibition was relatively robust for the influenza virus and SARS-CoV-2 and was mild for VSV-G. SARS-CoV-2, the causative pathogen of COVID-19, has rapidly spread worldwide, seriously endangering human health. Of note, nisoldipine was found to be capable of inhibiting SARS-CoV-2 and other human coronavirus OC43 (HCoV-OC43) infections in vitro without cytotoxicity (data unpublished). We deduce that nisoldipine displays broad-spectrum antiviral activity that may be effective against a variety of enveloped viruses. Therefore, whether nisoldipine inhibits other enveloped virus infections, especially by functionally similar modes of action, deserves further investigation.

Drug repurposing may be a rapid and powerful approach to counteracting SARS-CoV-2 and to ameliorating COVID-19 severity. Various studies have demonstrated the efficacy of repurposed drugs such as remdesivir, chloroquine and hydroxychloroquine for use against SARS-CoV-2 and as possible treatments for COVID-19 [45,46,47]. Notably, it is a critical issue that a loading dose would be specifically required against IAV infection. Considering that viral infection is an acute pathogenesis, there is a need to take higher doses of nisoldipine than those used in treating hypertension to achieve an optimal outcome. Thus, the administration route and the toxicity of a repurposed drug need to be carefully determined. Furthermore, these findings highlight not only the promise of nisoldipine as a repurposed drug against SARS-CoV-2, but also as a lead compound in developing new antivirals.

Ca^2+^influx through voltage-dependent calcium channels (VDCC) is an important pathway for the regulation of cytosolic Ca^2+^[48]. In cardiac tissues, CCBs inhibit calcium influx across the cell membrane of cardiac muscle and vascular smooth muscle by interfering with voltage-operated calcium channels. Ca^2+^ release mediates a number of downstream signals following virus particle entry, which in turn leads to efficient viral replication. Interference with intracellular calcium influx is critical to virus infection. It has been shown that the intracellular Ca^2+^ could regulate viral proteins or the function of cellular dependence factors to promote the internalization and replication of SFTSV (severe fever with thrombocytopenia syndrome virus) [49]. As nisoldipine is a CCB that mediates intracellular Ca^2+^ uptake, it can be speculated that the anti-viral effect is closely related to the intracellular Ca^2+^ level. Both BAPTA-AM and EGTA as specific calcium chelators could greatly diminish the relative intracellular IAV RNA level, suggesting that the intracellular Ca^2+^ level is important for IAV infection. The sialylation of Cav1.2 is critical for IAV infection, as previously documented [19,20]. To corroborate this finding, we transfected with small interfering RNAs (siRNAs) targeting Cav1.2 in the A549 cells. The knockdown of Cav1.2 largely inhibited and remarkably reduced viral NP levels, suggesting that IAV engages Cav1.2 for virus entry.

Using a fluorescent probe for the detection of calcium ion, we demonstrated that IAV infection was accompanied by an increase in the concentration of cytosolic Ca^2+^, which is consistent with Bao’s research [50]. They also found that IAV internalization required Ca^2+^ influx [50]. The treatment of A549 cells with nisoldipine could significantly reduce intracellular Ca^2+^. These data support our hypothesis that nisoldipine inhibits IAV infection by reducing the cellular Ca^2+^ level. Our results imply that nisoldipine inhibits IAV from entering host cells via Ca^2+^ influx–involved clathrin-mediated endocytosis.

Influenza viruses that are resistant to antiviral drugs frequently emerge [34]. The combination of anti-viral drugs with different mechanisms of action constitutes a promising approach to improving anti-viral activity via additive and synergistic effects. In particular, host-directed therapy is considered much less likely to result in the development of resistance. Zanamivir directly targeting viral NA enzymes is licensed for the treatment of influenza A and B infections. Our results showed that using a combination of nisoldipine and zanamivir displayed synergistic antiviral effects against IAV, highlighting a new use for old drugs and the combined use of existing antiviral drugs to hopefully become one of the most effective and rapid ways to fight IAV and other lethal acute viral infections.

In summary, we have shown that nisoldipine, an existing drug used for the treatment of hypertension, effectively inhibits the entry of IAV. The results of the mechanism study revealed that nisoldipine efficiently inhibited IAV entry through extracellular Ca^2+^ influx-involved clathrin-mediated and clathrin-independent endocytosis.

## 5. Conclusions

In conclusion, we have shown that nisoldipine, an existing drug used for the treatment of hypertension, effectively inhibits the entry of IAV without significant cytotoxicity. The subsequent mechanism study indicated that nisoldipine efficiently inhibited IAV entry through extracellular Ca^2+^ influx, involving clathrin-mediated and clathrin-independent endocytosis. This study shows that nisoldipine may be developed as a future therapeutic option for the treatment and prophylaxis of IAV infection.

## Figures and Tables

**Figure 1 viruses-14-02738-f001:**
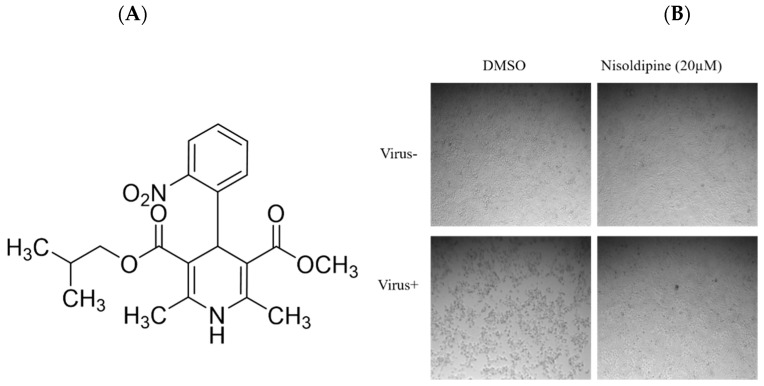
The effect of nisoldipine on the replication of IAV. (**A**) The chemical structure of nisoldipine. (**B**) Nisoldipine reduced virus-induced CPE in MDCK cells. Images of the cells were acquired at 4× magnification at 48 h post-infection. (**C**) Nisoldipine could dose-dependently reduce the formation of viral plaques. (**D**) The production load of progeny virus in the supernatant. The supernatants were diluted 1000 or 10,000 times, and virus titers were determined by means of plaque assays on MDCK cells. (**E**,**F**) The expression level of the viral protein and gene could also be restrained by nisoldipine in a dose-dependent manner in MDCK cells. (**G**,**H**) The expression level of the viral protein and gene could also be restrained by nisoldipine in a dose-dependent manner in A549 cells (* *p* < 0.05, ** *p* < 0.01, *** *p* < 0.001).

**Figure 2 viruses-14-02738-f002:**
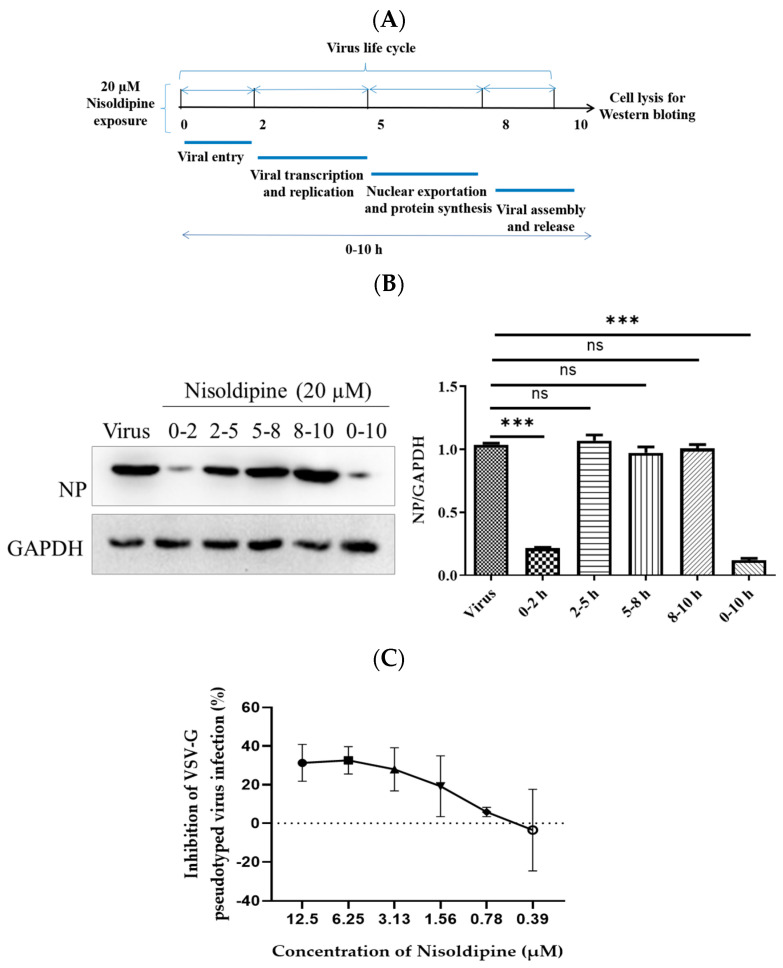
Nisoldipine blocked virus infection in the entry stage of IAV infection. (**A**) IAV−infected A549 cells were co−incubated with nisoldipine (20 µM) at different time points during a single replication cycle of the influenza A virus. Cell lysates were collected at 10 h after infection. (**B**) The expression of NP and GAPDH proteins was detected by Western blotting. (**C**) The inhibitory activity of nisoldipine against VSV−G PsV infection. Data are presented as mean ± SD (*** *p* < 0.001); ns means not significant.

**Figure 3 viruses-14-02738-f003:**
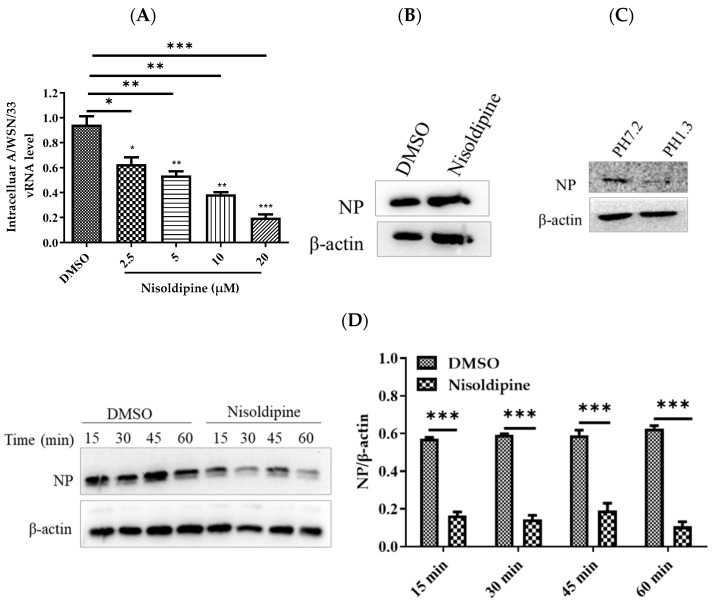
The effect of nisoldipine on viral internalization. (**A**) Nisoldipine acted in the pre-fusion phase of influenza A virus. A549 cells were treated with nisoldipine and then infected with IAV in the presence of ammonium chloride. At 24 h post infection, the cells were lysed to extract RNA, and the relative vRNA level of internalized IAV was measured. (**B**) Nisoldipine has not affected the adsorption of influenza A virus. A549 cells were pretreated with nisoldipine or DMSO for 3 h and then infected with IAV (MOI = 5) on ice at 4 °C for 1 h. The cells were lysed with RIPA and then subjected to Western blotting with a rabbit anti-NP pAb. (**C**) A549 cells were infected with A/WSN/33 (MOI = 5) on ice at 4 °C for 1 h, followed by ice-cold PBS (pH 7.2) or PBS-HCl (pH 1.3) before cell lysis. Viral NP protein expression was detected by Western blotting. (**D**) Nisoldipine could interfere with the internalization stage of the virus. A549 cells were pretreated with 20 μM nisoldipine or DMSO for 3 h and then infected with A/WSN/33 (MOI = 5) at 37 °C. The infected cells were washed with ice-cold PBS-HCl (pH 1.3) at the indicated time points before cell lysis. Viral NP protein expression was detected by Western blotting. Data are presented as mean ± SD (* *p* < 0.05, ** *p* < 0.01, *** *p* < 0.001).

**Figure 4 viruses-14-02738-f004:**
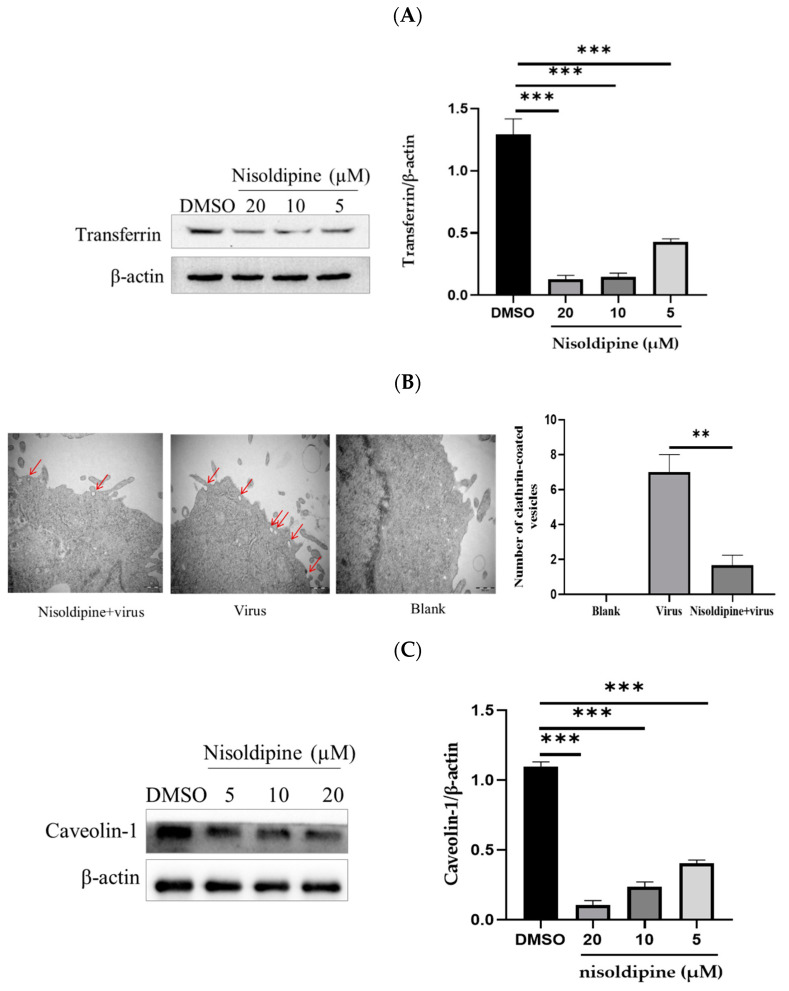
Nisoldipine inhibits endocytosis of IAV. (**A**) Nisoldipine could reduce the absorption of transferrin. A549 cells were pretreated with nisoldipine for 3 h and then incubated with transferrin (15 µg/mL) at 37 °C for 30 min. The cells were lysed with RIPA and then subjected to Western blotting with a rabbit anti-transferrin mAb. (**B**) Nisoldipine could reduce the number of clathrin-encapsulated endocytic vesicles. Scale bar: 500 nm. Magnification: 40,000×. Red arrows point to the clathrin-coated vesicle. (**C**) Nisoldipine could reduce the expression of caveolin-1. (**D**) Transferrin uptake was decreased when the cell was pretreated with nisoldipine. CPZ was used as a positive control. Scale bar: 20 μm. Tf-568 is shown in red. (**E**) The marked reduction of CTB endocytosis was observed in nisoldipine-treated cells. MβCD was used as a positive control. CTB-FITC is visualized in green. Data are presented as mean ± SD (** *p* < 0.01, *** *p* < 0.001).

**Figure 5 viruses-14-02738-f005:**
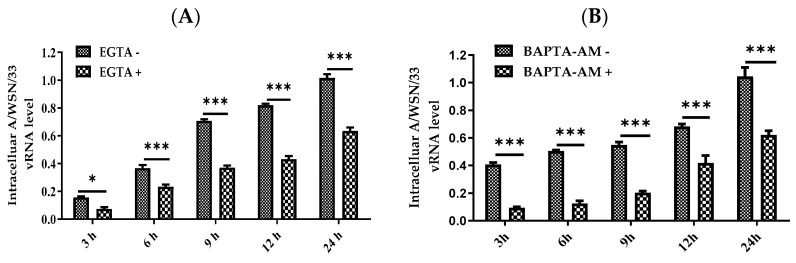
Nisoldipine inhibits IAV infection through reducing cellular Ca^2+^ uptake. (**A**,**B**) The relative level of vRNA in A549 cells upon treatment with EGTA or BAPTA-AM at 3, 6, 9, 12, 24 h post infection. (**C**) Intracellular Ca^2+^ concentration in the virus-infected cells. A549 cells were infected with influenza A/WSN/33 in the presence of nisoldipine. The level of intracellular Ca^2+^ was detected using fluorescence of BBcellProbeTMF3. (**D**) The viability of the cells after the knockdown of Cav1.2. (**E**) The protein levels of NP in the virus-infected A549 cells. Cells were transfected with siRNA targeting L-type calcium channel Cav1.2 for 24 h, followed by infected with A/WSN/33 for 12 h or 24 h. Cav1.2 protein and NP protein expression were detected by Western blotting. Data are presented as mean ± SD (* *p* < 0.05, *** *p* < 0.001); ns means not significant.

**Table 1 viruses-14-02738-t001:** Anti-influenza A virus activity of dihydropyridine calcium channel blockers (mean ± SD, *n* = 3).

Compound	Inhibition of A/Puerto Rico/8/34 Infection		
	IC_50_ (μM) ^a^	CC_50_ (μM) ^b^	SI (CC_50_/ IC_50_) ^c^
Nitrendipine	7.26 ± 1.89	171.05 ± 14.15	23.56
Nimodipine	12.83 ± 1.72	>200	>15.58
Cilnidipine	14.26 ± 2.66	>200	>14.02
Nisoldipine	4.74 ± 0.76	>200	>42.19
Zanamivir	1.26 ± 0.30 nM	>200 nM	>158.73

^a^ IC_50_: the half maximal inhibitory concentration. ^b^ CC_50_: the 50% cytotoxic concentration of compounds on the cells. ^c^ SI: selection index.

**Table 2 viruses-14-02738-t002:** Inhibitory activities of nisoldipine against multiple influenza virus strains (mean ± SD, *n* = 3).

IAV Strains	Inhibition Activity of Nisoldipine	
	IC_50_ (μM)	CC_50_ (μM)
A/Puerto Rico/8/34	4.74 ± 0.76	>200
A/FM-1/1/47	6.96 ± 0.78	
A/Aichi/2/68	5.68 ± 0.61	
A/WSN/33	4.47 ± 0.25	
A/PR/8/34 with NA-H274Y ^a^	5.94 ± 0.55	

^a^ Oseltamivir-resistant strain.

**Table 3 viruses-14-02738-t003:** Inhibitory activities of nisoldipine against different H5N1 pseudoviruses infection (mean ± SD, *n* = 3).

H5N1 Pseudovirus	IC_50_ (µM)
A/Qinghai/59/2005	5.84 ± 0.64
A/Xinjiang/1/2006	3.87 ± 0.51
A/Hong Kong/156/1997	4.47 ± 0.47
A/Anhui/1/2005	2.59 ± 0.46

**Table 4 viruses-14-02738-t004:** In vitro synergistic anti-influenza activity of nisoldipine with zanamivir.

Combination Ratio (IC_50_) ^a^	IC_50_ Equivalent ^b^		
Nisoldipine: Zanamivir	Nisoldipine	Zanamivir	^c^ FICI
7:1	0.61	0.09	0.70
3:1	0.49	0.17	0.66
1:1	0.27	0.27	0.54
1:3	0.12	0.37	0.49 ^d^
1:7	0.75	0.44	1.19

^a^ Combination ratio of nisoldipine is mixed with zanamivir in fixed ratios of concentrations corresponding to the IC_50_ equivalents of every single agent. ^b^ Concentration in IC_50_ equivalent was the normalized concentration that was calculated by dividing the IC_50_ of the drug in combination with its IC_50_ alone. ^c^ FICI was the sum of nisoldipine and zanamivir IC_50_ equivalent concentrations used in each combination. ^d^ The FICI was interpreted as synergism when the value was <0.5.

## Data Availability

The data presented in this study are available on request from the corresponding author.

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
