# Peer review of "Nisoldipine Inhibits Influenza A Virus Infection by Interfering with Virus Internalization Process"

_viruses, 2022, doi:10.3390/v14122738_

Round 1

Reviewer 1 Report

The manuscript entitled ‘Nisoldipine inhibits influenza A infection by interfering with virus internalization process’ characterizes the repurposing of a hypertension drug against influenza virus infection. The authors show in great detail the mechanism of nisoldipine inhibition of influenza virus replication by blocking internalization into host cells.

The authors state that they have previously performed a screen of compounds. Has this been previously published? If so, please cite.

The results associated with Table 1 state that they experiments were performed with WSN but the title of the table says PR8. Please clarify. Also, please define IC50, CC50 and SI in a table legend or in the text

Figure 1B is very blurry. Please include images with better resolution.

In Fig 2C, please adjust the x- and y-axes so that the difference is better appreciated and what does the solid line represent. Fig 2D, why do cells have a greater than 100% viability? Why does the graph in E have a line joining the points while the one is C doesn’t? For all figures please tell the reader what they are looking at in each graph and how many replicates or is an experiment representative of having done it ‘x’ amounts of times.

Can Figure 4B be quantified?

There were some spelling mistakes to be corrected. Acronyms like vRNA are used and then are spelled out (viral RNA) in the next sentence. Please be consistent. They authors state “Bao’s research” please provide a reference right after this.

Can the authors comment on the dosage that is used to treat hypertension versus the dosage that would be required to significantly block or inhibition viral infection?

Reviewer 2 Report

In this manuscript, the authors investigated the antiviral activity and mechanism of nisoldipine, which is a dihydropyridine calcium channel antagonist, against influenza A virus (IAV). They found that nisoldipine inhibited the entry of IAV into cells through extracellular Ca2+ influx associated with endocytosis. Also, they showed that nisoldipine may have an antiviral effect against SARS-CoV-2. The data presented in this manuscript possibly provide useful information on drug candidates for IAV. However, the overall quality (logic, presentation of the data, style, resolution of figures, etc.) should be largely improved. The followings are points that may improve the manuscript.

Major comments

1.         Throughout the manuscript, there are so many typographical and grammatical errors. The authors should consider asking someone (such as English native speakers and a professional editing service) to proofread their manuscript.

2.         Throughout the manuscript, the quality of the figure, such as resolution and style, seems too poor.

3.         Logical gaps are found throughout the manuscript.

4.         Page 6, Results, “We previously performed a screen of compound library of 361 ion channel inhibitors and identified candidate compounds with anti-IAV activity (inhibition rate>60%) without significant cytotoxicity at the concentration of 10 μM.”; This sentence should be supported by appropriate references. The citations here are very important to know the methods for primary screening, such as which compound library was used.

5.         Page 8, Results, “H5N1 pseudovirus”; Why do the authors suddenly talk about the H5N1 pseudovirus here? Based on the flow of the story, the authors should use WSN pseudovirus. If H5N1 pseudoviruses are used here, the authors should show the data from the experiment with the corresponding authentic viruses in section "3.1. In vitro Anti-influenza activity of Nisoldipine".

6.         Page 14, Discussion, “Excitingly, our results identified nisoldipine as a potential anti-SARS-CoV-2 candidate drug in vitro.”; This reviewer does not agree with this statement, because the authors showed the only IC50 and CC50 values using SARS-CoV-2 pseudovirus in the current version of manuscript. If the authors would like to describe nisoldipine as a potential candidate drug, inhibition mechanism against SARS-CoV-2 infection should be examined as well as IAV.

7.         Page 14, Discussion, “As reported in the recently pub-lished literature, SARS-CoV-2 might deposit their RNA genomes into the host cytoplasm through endocytosis[34], [35]. In the case of Ebola or Zika virus infection, viral entry is reported to require endocytic uptake[36], [37]. Considering the broad-spectrum antiviral potential of nisoldipine, it is noteworthy and deserves further investigation whether ni-soldipine other enveloped viruses with functionally similar modes of action.”; If nisoldipine is broadly effective against viruses that require endocytic uptake, how would you explain the VSV-G results in Figure 2C? In this respect too, the authors should investigate the inhibition mechanism against SARS-CoV-2 infection.

Minor comments

1.         Page 1, Title; “inhibits influenza A infection” should be “inhibits influenza A virus infection”

2.         Page 1, Title, “by interfering with virus internalization process”; According to page 14, 4th paragraph, the authors stated, “the internalization of IAV into late endosome were not affected”. But the authors did not describe “late endosome” at all. The authors should clarify these statements.

3.         Page 1, Abstract, “Influenza infections and recent SARS-CoV-2”; It's strange that the term “Influenza infections” and the term “recent SARS-CoV-2” are connected with the coordinating conjunction, “and”, because “SARS-CoV-2” is the name of virus.

4.         Page 1, Abstract, “SARS-CoV-2”; Spell out “SARS-CoV-2”.

5.         Page 1, Abstract, “urgent need”; This reviewer understands the “urgent need” for anti-SARS-CoV-2 drugs. However, do the authors think the development of anti-influenza drugs is also urgently needed?

6.         Page 1, Abstract, “new  antivirals”; Throughout the manuscript, there are several full-width spaces. Please correct them.

7.         Page 1, Introduction, “deaths[1]”; Throughout the manuscript, a half-width space should be inserted before the symbol "[".

8.         Page 2, Introduction, “Corona-virus pneumonia disease (COVID-19)”; Replace “Corona-virus pneumonia disease (COVID-19)” with “Coronavirus disease 2019 (COVID-19)”.

9.         Page 2, Introduction, “As yet, no drug has been shown to be sufficiently effective for treating COVID-19”; How about Pfizer’s paxlovid?

10.     Page 2, Materials and Methods, “Nisoldipine(98%”; Throughout the manuscript, a half-width space should be inserted before the symbol "(".

11.     Page 2, Materials and Methods, “BAPTA”; Spell out “BAPTA.”

12.     Page 2, Materials and Methods, “GAPDH”; Spell out “GAPDH.”

13.     Page 2, Materials and Methods, “HRP”; Spell out “HRP.”

14.     Page 2, Materials and Methods, “USA) “; Delete a half-width space.

15.     Page 3, Materials and Methods, “FBS”; Spell out “FBS.”

16.     Page 3, Materials and Methods, “A/FM-1/47/1”; “A/FM-1/47/1” should be ”A/FM-1/1/47.”

17.     Page 3, Materials and Methods, “A/WSN/1933”; The strain names “A/WSN/1933” and “A/WSN/3”3 are mixed in the manuscript. Should be unified to A/WSN/33 to match other strains. The authors should use A/WSN/33, because it's better to unify the digits of “year of collection” in the strain name.

18.     Page 3, Materials and Methods, “as describe previously”; Appropriate reference should be added.

19.     Page 3, Materials and Methods, “software”; Throughout the manuscript, the authors should add appropriate references for all software.

20.     Page 3, Materials and Methods, “RIPA”; Spell out “RIPA.”

21.     Page 4, Materials and Methods, “previously reported”; Appropriate reference should be added.

22.     Page 4, Materials and Methods, “as described previously.”; Appropriate reference should be added.

23.     Page 4, Materials and Methods, “as described previously.”; Appropriate reference should be added here, too.

24.     Page 4, Materials and Methods, “After three washes with ice-cold phosphate-buffered saline (PBS), the cells were lysed with RIPA and then subjected to Western blotting with a rabbit anti-NP pAb.”; The authors can use the abbreviation “PBS,” because PBS was already defined four lines above.

25.     Page 5, Materials and Methods, “fixed rations” and “ration of IC50”; Are the terms “ration” correct? Aren’t they “ratio”? Sorry if they are correct.

26.     Page 5, Materials and Methods, “FICI”; Spell out “FICI” somewhere.

27.     Page 6, Materials and Methods, “ns, no significant”; This should be noted in the figure legends.

28.     Page 6, Results, “The results showed that four candidate drugs had different inhibitory effects on Influenza A virus (A/WSN/33), and IC50 were 7.26±1.89 μM, 12.83±1.72 μM, 14.26±2.66 μM and 4.74±0.76 μM (Table 1), respectively.”; Which is correct, “Inhibition of A/Puerto Rico/8/34 infection” in Table 1 or “inhibitory effects on Influenza A virus (A/WSN/33)” in the text?

29.     Page 6, Results, “selection index”; The abbreviation “SI” should be added after the term “selection index.”

30.     Page 6, Results, “Fujioka y et al.”; Delete “y”.

31.     Page 6, Results, “including A/Puerto Rico/8/34,”; “(H1N1)” should be added after “A/Puerto Rico/8/34.”

32.     Page 6, Results, “We conducted the subsequent experiments with A/WSN/1933(H1N1) virus strain.”; Why did the authors used A/WSN/33 (H1N1) in the subsequent experiments? The authors should state the reason.

33.     Page 7, Results, “Figure 1”; In Figures 1C, D, and G, the concentrations of nisoldipine are shown as 20, 10, and 5 μM from left to right, but in Figures 1E, F, and H, they are reversed. It's easier to understand if they are in same order.

34.     Page 7, Results, “Figure 1A”; There is no description about Figure 1A in the text.

35.     Page 7, Results, “Figure 1D”; Why are there no data for 20 μM nisoldipine?

36.     Page 7, Results, “Figure 1D, F, and H”; In the method section, the authors stated, “In all cases, a P-value < 0.05 was regarded as statistically significant and marked with an asterisk (*).” But, it does not state the meaning of “***” and “****” in the Figure 1. Statistical significance should be explained in the figure legends. Moreover, for Figure 1F and H, is there a statistically significant difference only between virus and 20 μM nisoldipine? For example, how about between virus and 10 μM nisoldipine?

37.     Page 7, Results, “The supernatants were diluted 1000 or 10,000 times”; “1000” should be “1,000.”

38.     Page 7, Results, “Nisoldipine could inhibit viral proteins”; Is this statement correct? Does nisoldipine inhibit viral proteins?

39.     Page 8, Results, “CL-385319”; What does “CL-385319” mean? The explanation of CL-385319 should be added. Maybe a reference is needed.

40.     Page 8, Results, “we generated four H5N1 pseudovirus in 293T cells, including A/Qinghai/59/2005, A/Xinjiang/1/2006, A/Anhui/1/2005, and A/Hong Kong/156/1997.”; This should be written in the method section. Moreover, why were these strains selected? What about recent H5N1 strains?

41.     Page 8, Results, “nisoldipine has effect on viral entry”; In page 2, the authors stated, “Influenza viruses require the host cellular machinery for the virus life cycle which can be divided into virus attachment, entry, replication, and budding.” According to this definition, the possibility that nisoldipine has effect on virus attachment cannot be denied at this point.

42.     Page 8, Results, “We prepared SARS-CoV-2 pseudovirus according to a previously reported method”; Why do the authors suddenly talk about SARS-CoV-2 here? It is difficult to see the connection with the previous paragraph.

43.     Page 8, Results, “entry phase of SARS-CoV-2”; There is no explanation about the entry of SARS-CoV-2 throughout the manuscript. The explanation should be added in order to mention “entry phase of SARS-CoV-2” here.

44.     Page 8, Results, “Figure 2C”; VSVG should be VSV-G.

45.     Page 8, Results, “Figure 2C and E”; Spell out Psv somewhere.

46.     Page 9, Results, “Viral fusion”; Why do the authors suddenly talk about “viral fusion” here? It is hard to read the current version of manuscript due to the logical gap.

47.     Page 9, Results, “WSN”; “WSN” should be “A/WSN/33 (H1N1).” Or the abbreviation should be defined.

48.     Page 9, Results, “At 24 h post infection”; In the legend of Figure 3, it is described as “At 2 hpi.” Which is correct?

49.     Page 9, Results, “viral vRNA”; The term "viral vRNA" feels strange because “vRNA” itself stands for “viral RNA”.

50.     Page 9, Results, “Next, compound-treated A549 cells as described above were infected with IAV(MOI = 5)for 1 h on ice at 4°C.”; Does this IAV mean WSN? In the same section, it is easier to understand if the same term is used.

51.     Page 9, Results, “As shown in Figure 3D”; Figure 3D cannot be found.

52.     Page 10, Results, “Influenza exploits multiple endocytic pathways for infection.”; Replace “Influenza” with “Influenza virus”, because “Influenza” is a disease name.

53.     Page 10, Results, “The results showed the number of clathrin-coated vesicles decreased signifi-cantly when A549 cells were pretreated with nisoldipine.”; This sentence feels redundant. What do the authors think?

54.     Page 10, Results, “as previously reported”; Appropriate reference should be added.

55.     Page 11, Results, “Figures 4A and C”; As commented in Figure 1, in Figure 4A, the concentrations of nisoldipine are shown as 20, 10, and 5 μM from left to right, but in Figure 4C, they are reversed. It's easier to understand if they are in same order.

56.     Page 11, Results, “Figure 5”; The quality of figure feels too low. For example, the resolution of each figure is too low and the font sizes are too small. Moreover, same as the Figure 1, it does not state the meaning of “***” and “****”. Statistical significance should be explained.

57.     Page 13, Results, “Table 4. In Vitro Synergistic Anti-influenza Activity of nisoldipine with zanamivir.”; What is the criteria of capitalization?

58.     Page 13, Results, “Table 4”; The definition of “Combination ratio” is unclear. Please add an explanation clearly in the text.

59.     Page 13, Results, “Ddihydropyridines”; Ddihydropyridines” should be “Dihydropyridines”.

60.     Page 13, Discussion, “This latter group (Vera-pamil and diltiazem) has been shown to inhibit IAV infection (Nugent and Shanley, 1984)[19], [31].” For this latter group, is it known the inhibition mechanisms against IAV infection? If known, please compare and discuss with the mechanism reported in this study.

61.     Page 13, Discussion, “His274Tyr”; What is the reason for changing H274Y to His274Tyr in the same line?

62.     Page 14, Discussion, “nisoldipine at 20 μM”; What concentration is needed to treat hypertension? How does it compare to 20 μM? "Also, what about the side effects of nisoldipine?" Please have a discussion.

63.     Page 14, Discussion, “The severe acute respiratory syndrome-coronavirus type 2 (SARS-CoV-2)”; The term "SARS-CoV-2" has appeared many times before, so why spell it out here?

64.     Page 14, Discussion, “seriously endangering the human health seriously”; In this sentence, "seriously" appears twice.

65.     Page 14, Discussion, “Drug repurposing might be a fast and powerful approach to counteract SARS-CoV-2 and ameliorate COVID-19 severity.”; This reviewer agrees with this statement. Please discuss repurposed drugs for targeting SARS-CoV-2, such as emdesivir and chloroquine.

66.     Page 15, Discussion, “SFTSV”; Spell out “SFTSV”. In addition, this sentence should be supported by appropriate references.

67.     Page 15, Discussion, “Bao’s research”; Add a reference.

68.     Page 15, Discussion, “Influenza viruses resistant to antiviral drugs emerge frequently”; Add a reference.

69.     Page 15, Discussion, “host-directed therapy is considered much less likely to cause the development of resistance.”; This reviewer agrees with this statement. But, on the other hands, host-targeted antivirals might have side effects. What do the authors think about this issue?

Reviewer 3 Report

In this study, the authors repurposed Nisoldipine, a selective Ca2+ channel inhibitor commonly used to treat hypertension, as antiviral drug against influenza virus. The results form a preliminary investigation addressing the usage of this specific drug. The relevance and impact in more complex systems could have been addressed to provide more consolidated conclusions.

·         The authors should show the results related to SARS-CoV-2 if they are claiming it has any effect against SARS-CoV-2. Moreover, the authors have not explained a mechanistic hypothesis to use Nisoldipine as antiviral drug against SARS-CoV-2.

·         The evaluation of clathrin-mediated endocytosis could have been addressed quantitatively.

·         Mini-replicon assay is not adequate to evaluate viral entry

·         The authors could evaluate further the effect of the drug on viral entry using confocal microscopy also using endocytic markers.

·         Number references for drugs and antibodies should be included Materials and Methods sections, since the general name could be applicable to different products from the same company.

·         Supplementary data should also include validation of cell lines established by the group, such as 293T/ACE2.

·         Methods section does not include any reference to SARS-CoV-2 strain used pro the neutralization assay.
